# FoxP3^+^ Regulatory T-Cell Quantities in Nodal T-Follicular Helper Cell Lymphomas and Peripheral T-Cell Lymphomas Not Otherwise Specified and Their Impact on Overall Survival

**DOI:** 10.3390/cancers16234011

**Published:** 2024-11-29

**Authors:** Eva Erzar, Alexandar Tzankov, Janja Ocvirk, Biljana Grčar Kuzmanov, Lučka Boltežar, Veronika Kloboves Prevodnik, Gorana Gašljević

**Affiliations:** 1Department of Cytopathology, Institute of Oncology Ljubljana, Zaloška Cesta 2, 1000 Ljubljana, Slovenia; eerzar@onko-i.si (E.E.); vkloboves@onko-i.si (V.K.P.); 2Medical Faculty, University of Ljubljana, Vrazov Trg 2, 1000 Ljubljana, Slovenia; jocvirk@onko-i.si (J.O.); lboltezar@onko-i.si (L.B.); 3Pathology, Institute of Medical Genetics and Pathology, University Hospital Basel, University of Basel, Schönbeinstrasse 40, 4031 Basel, Switzerland; alexandar.tzankov@usb.ch; 4Division of Medical Oncology, Institute of Oncology Ljubljana, Zaloška Cesta 2, 1000 Ljubljana, Slovenia; 5Department of Pathology, Institute of Oncology Ljubljana, Zaloška Cesta 2, 1000 Ljubljana, Slovenia; bkuzmanov@onko-i.si; 6Medical Faculty, University of Maribor, Taborska Ulica 8, 2000 Maribor, Slovenia

**Keywords:** regulatory T cells, FoxP3^+^, peripheral T-cell lymphoma, prognostic biomarker, cut-off value, overall survival

## Abstract

The role of FoxP3^+^ regulatory T cells (Tregs) in the tumour microenvironment (TME) of peripheral T-cell lymphomas (PTCLs) is complex, and their impact on overall survival (OS) is unclear. This retrospective study aims to examine the quantity of FoxP3^+^ cells in the TME of PTCLs and reactive lymph nodes (LNs) and their impact on OS. A lower FoxP3^+^ cell quantity is found in PTCLs compared to reactive LNs. While differences in OS are observed between groups with high and low FoxP3^+^ cell quantities using various cut-off values, further analyses show no significant impact on the risk of death. This study suggests FoxP3^+^ cells as potential prognostic biomarkers but recommends the conduction of larger, multicentre studies with standardized protocols for confirmation. It also indicates that Treg-suppressing drugs may not be suitable for certain PTCL patients. Instead, combining therapies, including those enhancing Treg function, could be more effective in improving outcomes for PTCL patients, warranting further research.

## 1. Introduction

The tumour microenvironment (TME) plays an important role in the development and progression of cancer, particularly in relation to lymphomas [1,2,3]. Tumour-infiltrating immune cells (TICs) located in the TME are promising resources for the development of new prognostic biomarkers, as they are easily detectable [4,5]. One type of these promising immune cells comprise regulatory T cells (Tregs), a specialized subclass of CD4^+^ CD25^+^ T cells that express the transcription factor forkhead box P3 (FoxP3) [6,7]. These cells play an important role in the regulation of immune responses to self-antigens and nonself-antigens via local immune suppression, and may impair the anti-tumour response by reducing the activity and functions of T cells and natural killer (NK) cells [5,8].

The prognostic value of FoxP3^+^ Tregs has attracted substantial interest in recent decades. A high quantity of FoxP3^+^ Tregs is associated with decreased survival in most solid malignancies, such as breast cancer, ovarian cancer, and liver cancer [9,10,11]. In contrast, in relation to colorectal cancer [12,13,14,15] and different lymphoma subtypes [16,17,18,19,20,21,22], a higher quantity of FoxP3^+^ Tregs is linked to improved survival outcomes. These studies show that FoxP3^+^ Tregs may play a different role in lymphomas compared to solid tumours. Nevertheless, in specific lymphoma subtypes, increased levels of FoxP3^+^ Tregs have been associated with conflicting survival outcomes [16,18,23,24]. The reason for these differences may be, in part, that lymphoma subtypes are substantially heterogeneous and have diverse clinical and biological characteristics. Moreover, outcomes may also be influenced by factors such as the number of patients included in the study, treatment types, the use of different antibodies for immunohistochemical (IHC) staining, and the choice of cut-off values for defining patient groups with high or low quantities of FoxP3^+^ Tregs. 

In mature lymphomas, FoxP3^+^ Tregs constitute a significant component of the TME, and their quantity varies across B- and T-cell lymphoma (BCL and TCL) subtypes [16,18,19,20,22,24,25,26]. Their lower quantities within the TME of nodal T-follicular helper cell lymphomas and the presence of the angioimmunoblastic type (nTFHL-AI) compared to reactive (nonmalignant) lymph nodes (LNs) and follicular lymphomas (FLs) have been noticed [25], a fact which may partially account for the autoimmune symptoms commonly associated with nTFHL-AI and contribute to its unfavourable prognosis [23,25]. Notably, the presence of FoxP3^+^ Tregs within the TME of nTFHLs follicular type (nTFHLs-F), nTFHLs not otherwise specified (nTFHL-NOS), and composite lymphomas (CLs; nTFHL-AI and diffuse large B-cell lymphomas (DLBCLs) or marginal zone lymphomas (MZLs)) remains unexplored. 

The role of FoxP3^+^ cells in the TME of peripheral T-cell lymphomas (PTCL) is complex. This complexity stems, in part, from the fact that lymphoma cells are T cells and, thus, susceptible to potential inhibition by Tregs. Moreover, lymphoma cells could potentially originate from Tregs, as observed in adult T-lymphomas/leukemia (ATLL) which is defined by an etiological link to human T-cell lymphotropic virus type 1 (HTLV-1) infection and is the only subtype of PTCL characterized by a recurrent Treg phenotype [27]. The expression of FoxP3^+^ in lymphoma cells in peripheral T-cell lymphomas not otherwise specified (PTCLs-NOS) (HTLV-1 negative) is extremely rare. All published cases to date have been characterized by an extremely aggressive disease course and short survival [26,28,29,30]. Based on previous studies, it appears that a high quantity of FoxP3^+^ cells in the TME has a positive impact on survival outcomes in certain extranodal TCLs, such as mycosis fungoides (MF), unspecified cutaneous TCLs, and extranodal NK/T lymphomas (ENKTLs) [18,19,20,22]. In relation to nodal PTCLs, the knowledge about FoxP3^+^ cells in the TME and their impact on survival is limited. Only three studies have investigated their potential influence on survival, involving a small group of patients with anaplastic large cell lymphomas, nTFHLs-AI, and PTCLs-NOS [16,23,24]. The findings of Lundberg et al. did not confirm a noticeable impact of FoxP3^+^ cell quantity on overall survival (OS) [24]. In contrast, Tzankov et al. observed a suggestive, nonsignificant trend toward enhanced OS in nTFHL-AI patients with elevated FoxP3^+^ cell quantities [16], and Liu et al. showed a correlation between low numbers of FoxP3^+^ cells and a worse, progression-free survival (PFS), while OS was not statistically associated with FoxP3^+^ levels [23].

The aim of our study is to evaluate the quantity and prognostic value of FoxP3^+^ cells in the TME of 101 patients diagnosed with nTFHL-AI (all patterns), nTFHL-F, nTFHL-NOS, PTCL-NOS, and CL and compare them with the reactive LN patient cohort. This study finds fewer FoxP3^+^ cells in lymphoma subtypes such as nTFHL-F, advanced nTFHL-AI, and CL compared to reactive LNs. Although improved OS is observed in patients with a high FoxP3^+^ cell quantity, multivariate analysis did not confirm the quantity of FoxP3^+^ cells as an independent prognostic biomarker.

## 2. Materials and Methods

### 2.1. Patients and Study Design

Our retrospective study was performed on routinely collected histological samples. A total of 108 LN biopsies (whole-tissue samples) were obtained from lymphoma patients diagnosed with nTFHL-AI (*n* = 73), nTFHL-F (*n* = 4), nTFHL-NOS (*n* = 14), PTCL-NOS (*n* = 9), and CL, i.e., the co-occurrence of nTFHL-AI and DLBCL or MZL (*n* = 8). Moreover, 17 reactive LN biopsies were retrieved from the archives of the Institute of Oncology Ljubljana and various general hospitals across Slovenia. We eliminated three samples due to them containing an inadequate quantity of biological material, leaving us with 105 cases for analysis. All patients included in the study were diagnosed between 2007 and 2022 and had undergone treatment at the Institute of Oncology Ljubljana, Slovenia, in accordance with the valid clinical guidelines applicable at the time of diagnosis. Additional IHC staining on LN biopsies for samples lacking some of the necessary markers (Appendix A), as well as analyses on B- and T-cell clonality using the polymerase chain reaction (PCR) method (BIOMED-2) for samples lacking this information, were performed. The standardized and validated CE-IVD BIOMED-2 clonality assays (IdentiClone^®^ IGH Gene Clonality Assay and TCRB + TCRG T-Cell Clonality Assay-ABI Fluorescence Detection, InVivo Scribe Technologies, San Diego, CA, USA) were used according to the manufacturer’s instructions and according to the EuroClonality/BIOMED-2 guidelines for the interpretation and reporting of Ig/TCR clonality testing in suspected lymphoproliferation instances [31,32,33]. The target regions, primers used in the kit, and other details are described, in more detail, in the manufacturer’s product details. This allowed us to re-evaluate all initial diagnoses and subclassify nTFHLs-AI into patterns, in collaboration with experienced hematopathologists from the Institute of Clinical Genetics and Pathology, University Hospital Basel, Switzerland, according to the 5th edition of the World Health Organization Classification of Haematolymphoid Tumours: Lymphoid Neoplasms [34] and The International Consensus Classification of Mature Lymphoid Neoplasms [35]. Clinical data were obtained at the time of diagnosis or during patient follow-ups, and they were retrieved retrospectively from their medical records in our hospital information system. 

### 2.2. Immunohistochemical Staining for FoxP3^+^

From formalin-fixed paraffin-embedded (FFPE) tissue samples, 4 µm thick tissue sections were sliced. Automated IHC slide staining was performed on BenchMark Ultra IHC/ISH System (Roche Diagnostics, Ventana, Tucson, AZ, USA) using the OptiView DAB IHC Detection Kit ( Catal. No. 760-700, Roche Diagnostics, Ventana, Tucson, AZ, USA). A primary rabbit monoclonal antibody against FoxP3 (clone EP340, Epitomics portfolio, Cell Marque, Rocklin, CA, USA) was used and diluted 1:200 in a Dako REAL Antibody Diluent (Catal. No. S202230-2, Agilent Dako, Dako, Santa Clara, CA, USA). Two experienced hematopathologists (BGK and GG) independently evaluated IHC staining. First, the entire tissue section was scanned under 40× magnification, and the objective was then positioned in the area with the highest FoxP3^+^ cell density (hotspot area). In that area, visual scoring was performed by counting FoxP3^+^ cells under 400× magnification, with the hematopathologists blinded to all clinical data. Cells with moderate or strong staining were included in the count, and the result was reported as the number of FoxP3^+^ cells per square millimetre (FoxP3^+^ cells/mm^2^).

### 2.3. Statistical Methods

Descriptive statistics were used to summarize the clinical data. To determine if there were statistical differences in FoxP3^+^ cell quantities between the included nodal PTCL subtypes and the reactive LNs, a multiple comparison was carried out using the Kruskal–Wallis nonparametric test, and pairwise comparisons were performed using the Mann–Whitney U test. OS was defined as the time interval from the date of the pathological diagnosis to the date of death from any cause. The vital status of the patients was retrieved from the Cancer Registry of the Republic of Slovenia on 15 April 2024. The median survival time of the patients was expressed in months. The Kaplan–Meier method with a log-rank test (univariate analysis) was used to generate survival curves and evaluate the OS for patient groups with high or low FoxP3^+^ cell quantities. We used two different cut-off values from previous studies [19,24] (125 FoxP3^+^ cells/mm^2^ and 200 FoxP3^+^ cells/mm^2^), such studies investigating the impact on survival of FoxP3^+^ cells in different TCLs, and used the median count [18,22,23] of our cohort (255 FoxP3^+^ cells/mm^2^) to categorize patients into high or low FoxP3^+^ cell quantity groups. A potentially appropriate value that was suggested from our model (although the nonlinear effect was insignificant in our data) was also tested. The multivariate survival analysis was conducted using a Cox proportional hazards model (rms::cph()), with IPI as an ordered variable and a possibility of a nonlinear effect via restricted cubic splines for the Treg value (rms::rcs(), 3 knots). The resulting model was analyzed via the anova() function in base R, such a function estimating the significance of all components of the model, including the linear and nonlinear portions of the effect of the chosen predictor variable. Given the nonsignificance of the nonlinear effect, the results (which remain similar) show a simpler model where Treg is only included as a linear predictor. The proportional hazards assumption was tested using Schoenfeld residuals. A statistically significant result was defined at the *p* ≤ 0.05 level. The statistical analyses were conducted using IBM SPSS (v28.0.1.0), GraphPad Prism 9 (GraphPad software, San Diego, CA, USA), and R (v4.2.2) with RStudio (v2023.06.0+241) using the packages survival (v3.5-7), rms (v6.6-0), ggplot2 (v3.4.2), blandr (v0.5.1), knitr (v1.42), kableExtra (v1.3.4), and their dependencies.

## 3. Results

### 3.1. Subtypes of Nodal PTCL and Clinical Characteristics

Out of 108 nodal PTCL patients included in the study, three cases were excluded due to them containing inadequate biological material, and four cases of PTCL-NOS (HTLV-1-negative) were excluded due to the expression of FoxP3 in lymphoma cells. This resulted in a final cohort of 101 patients for analysis, comprising 72 individuals diagnosed with nTFHL-AI (nTFHL-AI pattern 1 (*n* = 7), nTFHL-AI pattern 2 (*n* = 34), and nTFHL-AI pattern 3 (*n* = 29); for two nTFHL-AI samples, we could not determine the pattern due to copious necrosis), 13 diagnosed with nTFHL-NOS, three diagnosed with nTFHL-F, five diagnosed with PTCL-NOS, and eight diagnosed with CL (as shown in Table 1). The analyzed patient cohort comprised fifty-six (55%) males and forty-five (45%) females, and the median age at diagnosis was 69 years (range 26–87 years). Most patients (*n* = 93; 92%) presented with advanced disease stages (Ann Arbor stages III–IV) upon diagnosis. Two-thirds presented with B-symptoms (*n* = 69; 68%), and a few more patients exhibited good eastern cooperative oncology group (ECOG) performance statuses (0–1) (*n* = 70; 70%) at diagnosis. In total, 13% of patients were in the low-risk IPI group, 22% in the low-intermediate group, 39% in the high-intermediate group, and 25% in the high-risk group. Elevated serum lactate dehydrogenase (LDH) was detected in 53 patients (53%). The main clinical characteristics are shown in Table 1. 

Approximately half of the patients (*n* = 58; 57%) received first-line treatments with chemotherapy, including cyclophosphamide, vincristine, prednisone (COP) or modified COP, or other low-dose treatments. Nineteen (19%) patients received chemotherapy with cyclophosphamide, doxorubicin, vincristine, and prednisone (CHOP) or CHOP-like treatments as the first-line treatment. Eighteen patients (18%) received other treatments (radiotherapy, corticosteroids, etc.), and the remaining six patients did not receive any treatment at all (*n* = 6; 6%). Patients with nTFHLs-AI, who made up the main group in this study, were treated with CHOP or a CHOP-like treatment in 16.7% of cases (*n* = 12) and with COP, modified COP, or other low-dose treatments in 66.7% of cases (*n* = 48), while 11.1% (*n* = 8) received other treatments and 5.6% (*n* = 4) did not receive any treatment. Autologous stem cell transplantation was only experienced by eight patients as a consolidation treatment after the initial therapy.

### 3.2. FoxP3^+^ Cell Quantity in nTFHLs, PTCLs-NOS, CLs, and Reactive LNs

FoxP3^+^ cells were detected in the TME of all analyzed samples (*n* = 101) in varying numbers (Figure 1), except for four PTCL-NOS cases (HTLV-1-negative) that expressed FoxP3 in the lymphoma cells (Figure 2). One sample contained 30–40% positive cells (P43), one sample contained weakly expressed FoxP3^+^ (P68), and the remaining two samples (P2 and P75) contained strongly expressed FoxP3^+^ in all lymphoma cells (Figure 2). These two patients had a very aggressive disease course and died within one month of diagnosis. 

The overall FoxP3^+^ cell quantity was significantly lower (*p* = 0.01) in the nodal PTCL cohort (median: 255 FoxP3^+^ cells/mm^2^; range: 4–2037 FoxP3^+^ cells/mm^2^) compared to the reactive LN cohort (median: 431 FoxP3^+^ cells/mm^2^; range: 199–791 FoxP3^+^ cells/mm^2^) (Figure 3). The FoxP3^+^ cell quantity did not vary across different nodal PTCL subtypes (Figure 4a), except for between nTFHLs-NOS and CLs (Kruskal–Wallis test: *p* = 0.02; Mann–Whitney U test: *p* < 0.05). However, FoxP3^+^ cell quantity exhibited a significant decrease in the nTFHL-F (median: 173 FoxP3^+^ cells/mm^2^; range: 79–177 FoxP3^+^ cells/mm^2^), nTFHL-AI (median: 279 FoxP3^+^ cells/mm^2^; range: 4–2037 FoxP3^+^ cells/mm^2^), and CL cohorts (median: 168.5 FoxP3^+^ cells/mm^2^; range: 117–403 FoxP3^+^ cells/mm^2^) compared to the reactive LN cohort (median: 431 FoxP3^+^ cells/mm^2^; range: 199–791 FoxP3^+^ cells/mm^2^) (Figure 4a). When we subclassified nTFHLs-AI into three patterns (patterns 1, 2, and 3), we observed differences compared to the reactive LNs (Figure 4b). The FoxP3^+^ cell quantity was statistically significantly reduced only in the pattern 3 cohort compared to the reactive LN cohort (*p* < 0.01), while the pattern 2 cohort nearly reached statistical significance (*p* = 0.06). Conversely, nTFHLs-AI pattern 1 had a median FoxP3^+^ cell quantity of 509 FoxP3^+^ cells/mm^2^ (range 23–1337 FoxP3^+^ cells/mm^2^) which did not differ from that of the reactive LNs (Figure 1 and Figure 4b) but was significantly higher (*p* < 0.05) compared to that of CLs (Figure 4b). When comparing patterns (Figure 4b), there was an almost statistically significant difference between patterns 1 and 3 (*p* = 0.05).

### 3.3. Survival Analysis Associated with FoxP3^+^ Cell Quantity

By the end of the follow-up period, 76 out of 101 patients had died (74.3%), and 64 had passed due to lymphomas. The median follow-up was 24.0 months. The median OS of the analyzed nodal PTCL cohort was 24.9 months (range 0.3–156.4 months). The 1-year, 2-year, and 5-year OS rates were 67.3%, 50.5%, and 23.8%, respectively. When comparing patients with low and high FoxP3^+^ cell quantities, determined using different cut-off values (125 FoxP3^+^ cells/mm^2^, 200 FoxP3^+^ cells/mm^2^, 255 FoxP3^+^ cells/mm^2^, or 400 FoxP3^+^ cells/mm^2^), we observed a statistically significant difference in OS between the two groups of patients when using the first three cut-off values (log-rank test; *p* < 0.001, *p* = 0.001, *p* = 0.004, and *p* = 0.066) (Figure 5). The Cox proportional hazards model showed a statistically significant association between IPI and OS (*p* < 0.001), no association between Treg values, on one hand, and diagnosis and OS, on the other (*p* = 0.35 and *p* = 0.92, respectively) (Table 2), and no nonlinear association between Treg values and OS (*p* = 0.44), as shown in Appendix A. However, due to the small sample size, the confidence interval of this estimate was broad, as shown in Figure 6. With a larger sample size, a significant association might be observed, in addition to a nonlinear association with a breaking point around 400 FoxP3^+^ cells/mm^2^. If this were to be observed in a more appropriate sample, further analysis of such a breaking point would be warranted, with the possibility of using it as a cut-off value between high-risk and low-risk patients. 

## 4. Discussion

TICs are an ideal source for the development of novel prognostic biomarkers because they are easily detectable. The prognostic potential of FoxP3^+^ Tregs and their varying numbers in the TME has attracted substantial interest, especially in relation to different lymphoma subtypes from which positive and negative prognostic effects have been reported [5,16,18,22,23,24]. Currently, limited data are available on the FoxP3^+^ cell quantity and their prognostic value in nodal PTCLs. 

In this study, we assessed FoxP3^+^ cell quantities in the TME of 101 nodal PTCLs and in 17 reactive LNs and analyzed the impact of FoxP3^+^ cell quantity on OS. Statistically lower FoxP3^+^ cell quantities in the nodal PTCL cohort compared to the reactive LN cohort were observed. Importantly, we are the first to demonstrate a statistically significant difference in OS among nodal PTCLs based on FoxP3^+^ cell quantity using various cut-off values obtained from previous publications [18,19,22,23,24]. However, the Cox proportional hazards model did not indicate a significant correlation between FoxP3^+^ cells (Treg value) and the risk of death.

A median quantity of FoxP3^+^ cells (median: 255 FoxP3^+^ cells/mm^2^; range: 4–2037 FoxP3^+^ cells/mm^2^) similar to those reported by Lundberg et al. [24] and Kim et al. [19] was obtained in our cohort. Lundberg et al. reported a median of 342 FoxP3^+^ cells/mm^2^, with a similarly wide range (1–3047 FoxP3^+^ cells/mm^2^) in their cohort of various TCL subtypes (*n* = 35) [24]. Kim et al. found a median of 205 FoxP3^+^ cells/mm^2^ (with a range of 0–3743 FoxP3^+^ cells/mm^2^) in their cohort of ENKTL cases [19]. This study found a relatively higher mean count of FoxP3^+^ cells/mm^2^ compared to that reported by Tzankov et al. in both nTFHL-AI (387 vs. 61) and PTCL-NOS (301 vs. 34) cohorts [16]. We did not detect a statistically significant difference in the quantity of FoxP3^+^ cells among the TME of the included nodal PTCL subtypes, except for between nTFHLs-NOS (median: 408 FoxP3^+^ cells/mm^2^; range: 78–1178 FoxP3^+^ cells/mm^2^) and CLs, i.e., the co-occurrence of nTFHLs-AI and BCLs (median: 186.5 FoxP3^+^ cells/mm^2^; range: 117–403 FoxP3^+^ cells/mm^2^) (Figure 4a). Similarly to our findings, both groups studying nTFHLs-AI and PTCLs-NOS [16,24] reported a higher median quantity of FoxP3^+^ cells in nTFHL-AI patients compared to PTCL-NOS patients, but this difference was not statistically significant across the three studies. However, we did not observe this trend in the nTFHL-AI cohort. When nTFHLs-AI were subclassified into patterns (1, 2, and 3), a similar nonsignificant trend was observed in the nTFHL-AI pattern 1 but not in the other two patterns (Figure 4b). Moreover, we noted higher FoxP3^+^ cell quantities in the nTFHL-AI pattern 1 cohort compared to the CL cohort (*p* ≤ 0.05). Our findings suggest the existence of a trend where the quantity of FoxP3^+^ cells decreases as nTFHLs-AI progress from pattern 1 to pattern 3, the most advanced stage of disease development, and, ultimately, to CL, where secondary BCL develops (Figure 1 and Figure 4b). Secondary lymphomas only develop in the presence of nTFHLs-AI pattern 3, while pattern 1 represents an indolent stage of nTFHL-AI that is not associated with secondary BCL development, as reported by Tan et al. [36]. This may help explain our findings, which are consistent with those reported by Gjerdrum et al. for cutaneous T-cell lymphomas (CTCL) [18]. They observed higher numbers of FoxP3^+^ Tregs in patients presenting with earlier stages of MF compared to patients suffering from advanced MF and presenting tumours or transformations into large cell lymphomas. Similar findings were also observed in FLs by Carreras et al. [17]. Based on our observations, we hypothesize that more FoxP3^+^ cells are present in the TME of less advanced stages of nTFHLs-AI. Further validation of this trend requires a larger sample size for statistical confirmation.

In our study, FoxP3^+^ cell quantities were significantly lower in the nodal PTCL cohort compared to the reactive LN cohort (Figure 3). The quantity of FoxP3^+^ cells was significantly reduced in the nTFHL-F (median: 173 FoxP3^+^ cells/mm^2^; range: 79–177 FoxP3^+^ cells/mm^2^), nTFHL-AI (median: 279 FoxP3^+^ cells/mm^2^; range: 4–2037 FoxP3^+^ cells/mm^2^), and CL (median: 168.5 FoxP3^+^ cells/mm^2^; range: 117–403 FoxP3^+^ cells/mm^2^) subtypes compared to reactive LNs, with 431 FoxP3^+^ cells/mm^2^ (range: 199–791 FoxP3^+^ cells/mm^2^, Figure 4a). Similarly, Bruneau et al. showed that nTFHLs-AI are associated with reduced FoxP3^+^ cell quantities compared to reactive LNs and FLs [25]. However, when we analyzed nTFHL-AI patterns separately, the quantity was not reduced in the nTFHL-AI pattern 1 cohort (*p* = 0.49; median: 509 FoxP3^+^ cells/mm^2^; range: 23–1337 FoxP3^+^ cells/mm^2^) and in the nTFHL-AI pattern 2 cohort (*p* = 0.06; median: 268 FoxP3^+^ cells/mm^2^; range: 28–2037 FoxP3^+^ cells/mm^2^) compared to the reactive LN cohort (Figure 4b), possibly due to one large outlier (with 2037 FoxP3^+^ cells/mm^2^). This patient was still alive at the time of writing despite having been diagnosed in 2011, a phenomenon which may be partially attributable to the high FoxP3^+^ cell quantity. We attribute the lack of decreases in pattern 1 to the initial histological stage of the disease. In contrast, the most significant decrease in FoxP3^+^ cell quantities in CLs compared to reactive LNs (*p* ≤ 0.001) was attributed to the simultaneous presence of two distinct types of lymphomas (nTFHLs-AI and BCLs). 

Since nTFHLs are not derived from Tregs, unlike ATLL and some rare PTCL-NOS cases, the lymphoma cells in these subtypes are not expected to express FoxP3. Accordingly, we detected FoxP3 expression only in the lymphoma cells from four HTLV-1-negative PTCL-NOS cases (Figure 2). In addition, the nuclei of FoxP3^+^-reactive T cells were substantially smaller and more common than those of the lymphoma cells. FoxP3 expression in PTCL-NOS lymphoma cells is extremely rare, with few published cases showing aggressive progression and death shortly after diagnosis [26,28,29,30]. We found two cases with strong FoxP3 expression in the lymphoma cells, and this is consistent with the clinical presentation observed in published cases [26,29,30]. The two other cases exhibited weak FoxP3 expression or expression in 30–40% of the lymphoma cells, both showing relatively short survival times.

Despite the potential of TICs as prognostic and predictive biomarkers, setting their cut-off values poses a challenge. The cut-off value serves as a basis for dividing patients into two groups (high vs. low cell quantity) and for evaluating their impact on OS. Different cut-off values, alongside other factors such as small sample sizes, can affect the final outcome of various research groups and their subsequent comparisons. Currently, cut-off values for the number of FoxP3^+^ cells are not standardized across published articles that study FoxP3^+^ cells and their impact on OS [16,17,18,19,20,23,24], and results are reported using different units, including percentages (%), cells per high-power field (cells/HPF), and cells per square millimetre (cells/mm^2^). To be able to compare our results with the studies published thus far, we decided to use the same cut-off values that had been used in previous publications [18,19,22,23,24]. We did not include the cut-off value of Tzankov et al. because they found substantially lower overall FoxP3^+^ cell quantities compared to those reported by Lam et al. and our study; this is because we could not convert their unit into ours [16,19]. In these TCL studies, researchers used three methods to determine the cut-off values: arbitrarily defined [24], median cohort counts [18,22,23], and ROC curve-derived values [16,19,20]. However, it is important to note that the latter approach is less suitable for studies with smaller sample sizes, a challenge that is often encountered when researching rare diseases such as PTCL. We also tried to determine the statistically appropriate cut-off value by testing the nonlinearity effect in our data. A potentially appropriate cut-off value could be around 400 FoxP3^+^ cells/mm^2^, as suggested by our model (Figure 6), if the nonlinear effects were significant in relation to our data. To enable meaningful cross-study comparisons, it is important for the scientific community to establish standardized cut-off values and units for TICs. 

When comparing patients with low and high FoxP3^+^ cell quantities, determined using different cut-off values (125 FoxP3^+^ cells/mm^2^, 200 FoxP3^+^ cells/mm^2^, 255 FoxP3^+^ cells/mm^2^, or 400 FoxP3^+^ cells/mm^2^), we observed a significant difference in OS (univariate analysis) between the two groups, except for the cases where a cut-off value of 400 FoxP3^+^ cell/mm^2^ was used (Figure 5d); this was expected, as the nonlinear effect was insignificant in our data (Appendix A). Unlike us, Lundberg et al. did not observe a significant difference in OS when using a cut-off value of 200 FoxP3^+^ cells/mm^2^, but they included significantly fewer TCL patients (*n* = 35) in their study [24]. Moreover, Liu et al. did not observe a significant difference in OS when using a median count of all the analyzed nTFHL-AI samples (*n* = 46) as a cut-off value [23]. Tzankov et al. observed a nonsignificant trend toward better OS in the nTFHL-AI cohort when using ROC curve-derived cut-off values [16]. The four groups that were studied with respect to FoxP3^+^ cell quantities in the CTCL and ENKTL cohorts exhibited prolonged OS in relation to patients with increased FoxP3^+^ cell quantities [18,19,20,22]. Our Cox proportional hazards model (multivariate analysis) showed no association between FoxP3^+^ cell quantity (Treg value) and the risk of death (*p* = 0.35) and no nonlinear association between Treg and the risk of death (*p* = 0.44). A decreasing risk value was found as the Treg value fell toward 400 FoxP3^+^ cells/mm^2^, and then a constant risk emerged at Treg values equal to or greater than 400 FoxP3^+^ cells/mm^2^ (Figure 6). Unfortunately, we cannot draw reliable conclusions about this phenomenon, as the Treg value was not statistically and significantly associated with the risk of death, and the indicated nonlinear effect was also statistically insignificant. With a larger sample size, a significant association might be observed, in addition to a nonlinear association with a breaking point (cut-off value) around 400 FoxP3^+^ cells/mm^2^. If this were to be confirmed in a larger, more appropriate sample, further analysis on this breaking point would be justified, with the potential to use it as a cut-off value to distinguish between high-risk and low-risk patients. According to the results of the multivariate analysis, only IPI was identified as an independent prognostic marker (*p* < 0.001). Lundberg et al. and Tzankov et al. also found no association between the Treg value and the risk of death through multivariate analysis [16,24], while Gjerdrum et al. and Kim et al. confirmed the opposite, but for cutaneous T-cell lymphomas and ENKTLs [18,19]. However, these studies either used the quantity of FoxP3^+^ cells as a categorical variable or failed to specify whether it was treated as a continuous or categorical variable based on their determination of the cut-off values. This ambiguity renders the comparison of their results with ours challenging, and drawing any definitive conclusions is difficult. The median OS for all patients in our nodal PTCL cohort was 24.9 months (range 0.3–156.4 months). The 1-year, 2-year, and 5-year OS rates were 67.3%, 50.5%, and 23.8%, respectively. Liu et al. reported similar 2-year OS (50.5%), 5-year OS (27.2%), and median OS (24.2 months; range 0.6–156.2 months) [23], while Lundberg et al. found a higher 1-year OS (89%) and 5-year OS (46%) [24].

Our study has a few limitations. Firstly, it is a retrospective study using registry data. Secondly, with the small patient groups representative of certain nodal PTCL subtypes, the study may lack sufficient power to detect a statistically significant difference among them. The smaller the group, the more variability is introduced, making it harder to distinguish between true effects and random noise. Smaller groups tend to exhibit more variability, with individual differences or outliers potentially skewing the results and resulting in unreliable conclusions. Another limitation could be the potential geographic or institutional biases arising from differences in sample handling and immunohistochemistry, possibly restricting the generalization of this study’s findings. Additionally, using only the FoxP3 marker may include heterogeneous lymphocyte subpopulations. However, we benefited from using whole-tissue sections and immunohistochemistry, both of which are cost-effective and simple to use in clinical studies and routine practice.

This study reveals a decrease in FoxP3^+^ cell quantities in nTFHLs-F, the advanced phase of nTFHLs-AI (pattern 3), and in CL subtypes compared to reactive LNs. This marks the first demonstration of such a difference among nTFHL-AI patterns. We illustrate the distinction in OS (in univariate analysis) between the groups with low and high FoxP3^+^ cell quantities by applying various cut-off values. However, our multivariate analysis did not confirm the role of the Treg value as an independent prognostic biomarker. To validate our findings, we suggest that future prospective studies utilize even larger sample sizes to obtain greater statistical power, employ consistent therapy protocols, use uniform detection antibodies with additional T-cell markers to include more homogeneous lymphocyte subpopulations, and adopt standardized methodologies and criteria for cut-off value determination and measurement unit selection. This rigorous approach could strengthen future research and is essential for determining the appropriate cut-off value and confirming the prognostic value of FoxP3^+^ cells in the TME of nodal PTCLs.

## 5. Conclusions

Our findings suggest that FoxP3^+^ cells could serve as potential prognostic biomarkers, but larger, multicenter studies relying on standardized protocols are needed for confirmation. Our study points to the possibility that Treg-suppressing drugs, such as cyclosporine, may not be suitable for use in nodal PTCL patients with a reduced quantity of FoxP3^+^ cells. In contrast, the possibility of treating nodal PTCLs with combined therapies, including immunomodulatory drugs that enhance Treg quantity and function, could offer a promising strategy for future treatment. However, these therapeutic approaches need further validation through the acquisition of more robust findings in future clinical studies. Such combined strategies have the potential to better modulate the TME and improve PTCL patients’ outcomes.

## Figures and Tables

**Figure 1 cancers-16-04011-f001:**
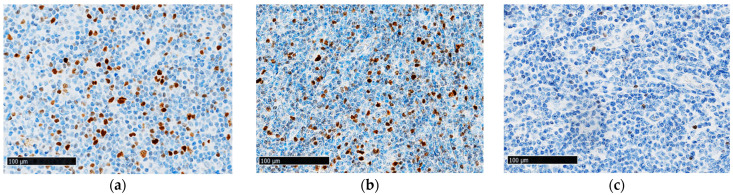
Representative images of FoxP3^+^ cells in (**a**) reactive lymph nodes (LNs), (**b**) in the tumour microenvironment (TME) of nTFHLs-AI pattern 1, and (**c**) in the TME of nTFHLs-AI pattern 3. Scale bar = 100 µm.

**Figure 2 cancers-16-04011-f002:**
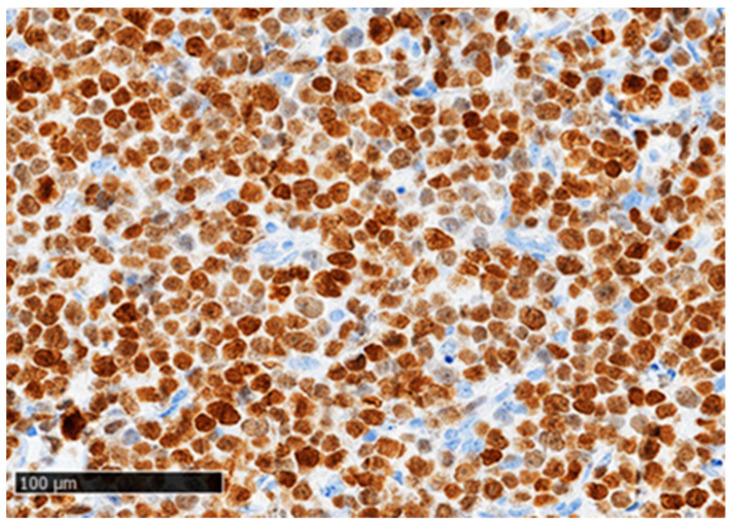
FoxP3 expression in the lymphoma cells in a human T-cell lymphotropic virus type 1 (HTLV-1) negative PTCL-NOS sample (P2-patients’ ID number) under 200× magnification. Scale bar= 100 µm.

**Figure 3 cancers-16-04011-f003:**
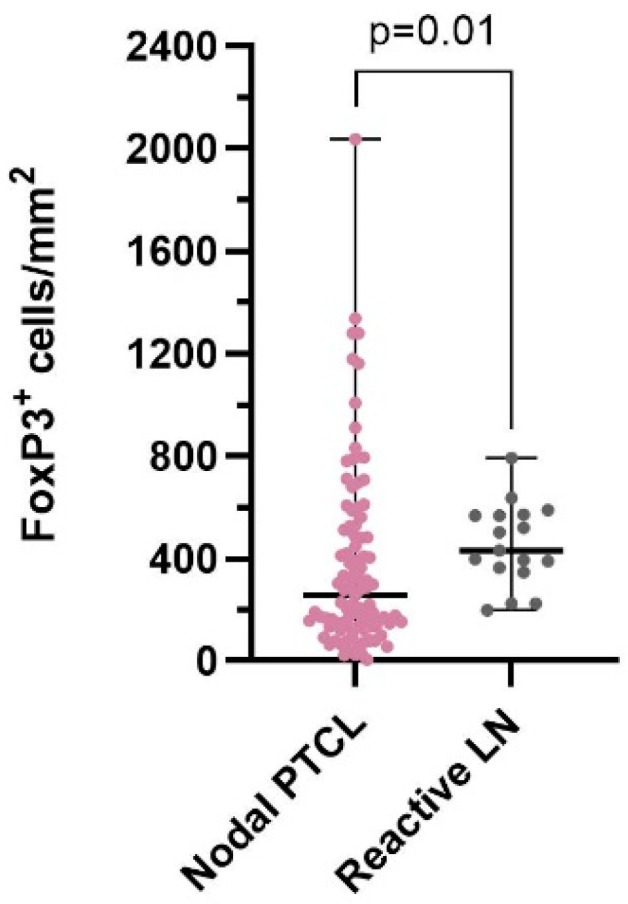
Comparison of FoxP3^+^ cell quantities (median with range) between the nodal peripheral T-cell lymphoma (PTCL) cohort and the reactive LN cohort.

**Figure 4 cancers-16-04011-f004:**
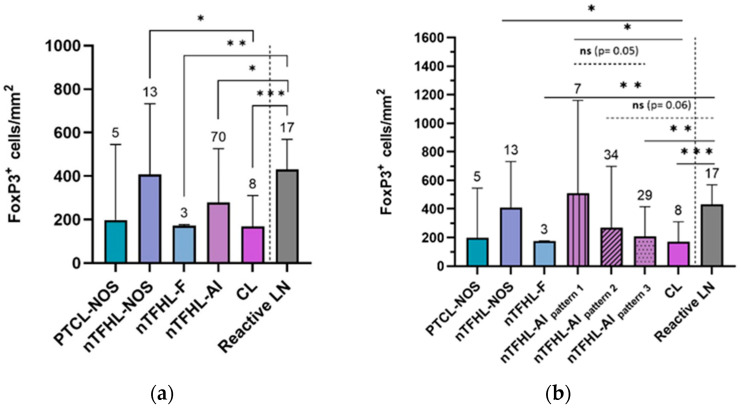
(**a**) Quantity of FoxP3^+^ cells (median with interquartile range) in the TME of different nodal PTCL subtypes and reactive LNs; (**b**) different nodal PTCL subtypes, with nTFHLs-AI divided into three patterns, and reactive LNs. Degrees of statistical significance: * *p* < 0.05; ** *p* < 0.01, *** *p* < 0.001, and ns (nonsignificant).

**Figure 5 cancers-16-04011-f005:**
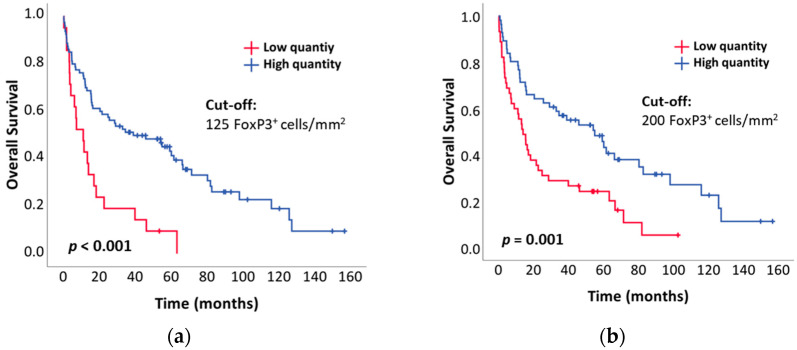
Comparison of the overall survival (OS) of all patients included in the study with high vs. low FoxP3^+^ cell quantities: (**a**) cut-off value of 125 FoxP3^+^ cells/mm^2^ based on Kim et al. [19]; (**b**) cut-off value of 200 FoxP3^+^ cells/mm^2^ based on Lundberg et al. [24]; (**c**) cut-off value of 255 FoxP3^+^ cells/mm^2^ based on the median count [18,22,23] of all the analyzed samples from our cohort; (**d**) cut-off value of 400 FoxP3^+^ cells/mm^2^ as suggested by the testing for the nonlinear effects in our data. The *p*-values of statistically significant differences are provided alongside the *p*-values of ns differences.

**Figure 6 cancers-16-04011-f006:**
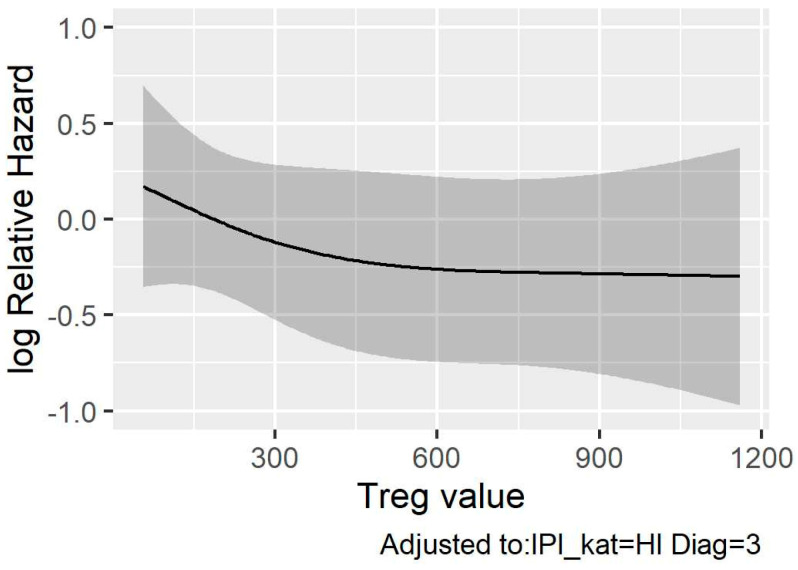
Association between OS (logarithm of the relative hazard of death) and the Treg value (as a continuous variable) if the nonlinear effect were to be allowed in the model, with adjustments for diagnosis and IPI. The gray shaded area indicates the 95% confidence interval.

**Table 1 cancers-16-04011-t001:** Main clinical characteristics and treatments received by the patient group included in the analyses.

Characteristics	No. of Patients (%)
Total	101 (100)
**Diagnosis**	
PTCL-NOS	5 (5)
nTFHL-NOS	13 (13)
nTFHL-F	3 (3)
nTFHL-AI	72 (71)
CL (nTFHL-AI and DLBCL or MZL)	8 (8)
**Gender**	
Male	56 (55)
Female	45 (45)
**Age at diagnosis**	
>60	79 (78)
≤60	22 (22)
**Ann Arbor Stage**	
I	1 (1)
II	7 (7)
III	24 (24)
IV	69 (68)
**B-symptoms**	
No	32 (32)
Yes	69 (68)
**Raised serum LDH level**	
No	46 (46)
Yes	53 (53)
Not available/unknown	2 (2)
**ECOG performance status**	
0	35 (35)
1	35 (35)
2	19 (19)
3	5 (5)
4	7 (7)
**IPI risk groups**	
Low-risk group (0, 1)	13 (13)
Low-intermediate-risk group (2)	22 (22)
High-intermediate-risk group (3)	39 (39)
High-risk group (4, 5)	25 (25)
Not available/unknown	2 (2)
**First-line treatment**	
CHOP/CHOP-like	19 (19)
COP/modified COP/other low-dose treatments	58 (57)
Other	18 (18)
None	6 (6)
**Death**	
Yes	26 (26)
No	75 (74)

PTCL-NOS: peripheral T-cell lymphoma not otherwise specified; nTFHL-NOS: nodal T-follicular helper cell lymphoma not otherwise specified; nTFHL-F: nodal T-follicular helper cell lymphoma, follicular type; nTFHL-AI: nodal T-follicular helper cell lymphoma, angioimmunoblastic type; CL: composite lymphoma, co-occurrence of nTFHL-AI and diffuse large B-cell lymphoma (DLBCL) or marginal zone lymphoma (MZL); LDH: lactate dehydrogenase; ECOG: eastern cooperative oncology group; IPI: international prognostic index.

**Table 2 cancers-16-04011-t002:** Results of a Cox proportional hazards model (multivariate analysis) including the Treg value as a linear variable.

Variables Included	HR [95% CI]	*p*-Value
**Diagnosis**		0.92
nTFHL-NOS	1.25 [0.35–4.42]	
nTFHL-AI	1.48 [0.55–4.01]	
Composite lymphoma (CL)	1.30 [0.38–4.41]	
nTFHL-F	1.80 [0.38–8.6]	
**IPI risk groups**		<0.001
Low-intermediate	1.06 [0.39–2.89]	
High-intermediate	1.98 [0.79–4.94]	
High	7.58 [2.82–20.37]	
**Treg value (50 units)**	0.98 [0.95–1.02]	0.35

## Data Availability

Due to ethical and privacy reasons, the data presented in this study are only available upon request from the corresponding author.

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
