# Peer review of "FoxP3+ Regulatory T-Cell Quantities in Nodal T-Follicular Helper Cell Lymphomas and Peripheral T-Cell Lymphomas Not Otherwise Specified and Their Impact on Overall Survival"

_cancers, 2024, doi:10.3390/cancers16234011_

Round 1
Reviewer 1 Report
Comments and Suggestions for Authors
The authors of the manuscript " " investigated the amount of FoxP3+ Tregs in different lymphomas. High quantity of FoxP3+ Tregs is associated with decreased survival in most solid tumors.
whereas an increased level of FoxP3+ Tregs in lymphomas is linked to improved survival outcomes.
The aim of our study was to evaluate and compare the quantity and the prognostic value of FoxP3+ cells in TME among 101 patients diagnosed with nTFHL-AI (all patterns), 104 nTFHL-F, nTFHL-NOS, PTCL-NOS, and CL, as well as those with reactive LN.
Although the study sounds interesting it still needs some improvement.
First of all the patient groups are to0 small to draw any conclusions. In particular PTCL-NOS, nTFHL-F and CL.
There is no clear result for the power of FoxP3+ Tregs as a prognostic marker. Furthermore it is not clear, if there is any different effect in the different cell lines.
The overall survival in the patient cohort is increased with increased number of FoxP3+ Tregs; this is not really new.
Finally, the results of the the authors do not support their conclusion.
The study needs more patient samples to support the idea that FoxP3+ Tregs can be used as diagnostic marker for the different lymphoma types.
Comments on the Quality of English LanguageThe text ist difficult to read and has to be improved to be more clear on the topic.
Author Response
"Please see the attachment."

Reviewer 2 Report
Comments and Suggestions for Authors
The original clinical article deals with FoxP3+ expressing T-regulatory cells in peripheral T- cell lymphomas and their impact on overall survival of the patients. Due to rarity of the disorder, this issue is only scarcely studied, especially, in view of steadily changing lymphoma classification and therapies. The authors have collected the quite representative group of adult patients thus enabling distinct general conclusions confirming some previous studies in the field.
Remarks:
Line 74: nodal T-follicular helper cell lymphoma, angioimmunoblastic type (nTFHL-AI) comprised the main group of the patients under study, as seen from Table 1. Therefore, the treatment protocols for this clinical form should be described in more details. Moreover, the modifications of TFHL low-dose therapy over a decade of studies could be mentioned.
Line 120: More details should be given on IHC and/or PCR (?) assessment of B- and T-clonality (Commercial kit? What genes were targeted if PCR assay was used? Manufacturer or home-made probes should be specified?)
Line 115: Which clinical/laboratory criteria were used for the diagnosis of reactive lymphadenopathy, to exclude neoplastic or viral pathology?
Line 118: Due to continuous recruiting period (2007 to 2022), which was the median follow-up period?
Line 125: A reference should be provided for the WHO classification in the reference list.
Results:
Table 1: Please provide median age of the patients (min/max range).
What number of patients were tested for HTLV persistence (blood or biopsy testing)?
Line 249: how were assessed the cutoff values for FoxP3 cells amounts (ROC analysis?) – it should be specified in Methods.
Conclusions are rather modest, but well justified. Limitations of the study are mentioned in Discussion.
Minimal copy checking is required.
Author Response
"Please see the attachment."

Reviewer 3 Report
Comments and Suggestions for Authors
Cancers (ISSN 2072-6694)
Manuscript ID
cancers-3258119
FoxP3+ Regulatory T-Cell Quantity in Nodal T-Follicular Helper Cell Lymphomas and Peripheral T-cell Lymphomas, Not Otherwise Specified and Its Impact on Overall Survival
Eva Erzar , Alexandar Tzankov , Janja Ocvirk , Biljana Grčar Kuzmanov , Lučka Boltežar , Veronika Kloboves Prevodnik , Gorana Gašljević *
Regulatory T (Treg) cells expressing the transcription factor forkhead box P3 (Foxp3), namely FoxP3+ regulatory T-cells, are important in body immune surveillance that could be potentially a therapeutic tool by reprogramming these cells to alter immune environments to cancer cells so as to cure cancer. The exact role of FoxP3+ cells in both nodal TFH cell lymphoma and PTCL, NOS is illusive and their impact on disease prognosis and survival is understudied.
The study aimed to measure the quantity of FoxP3+ cells in nodal PTCLs (n=105) and reactive lymph nodes (LNs) (n=17) and compare their impact on OS. The results emphasized the key role of FoxP3+ cell quantity as one of prognostic factors and highlighted potential use of Treg-stimulating therapies in PTCLs. It sounds an interesting topic and potential beneficial to clinical treatment. However, the study is limited in size of control specimens (reactive lymph nodes) and contains a heterogenous group of PTCL diagnosis, which could lead to interpretation bias. It is recommended to increase sample size in control group and reorganize the data before consideration for publication.
Critiques
1. Methods and materials: it is unclear to readers how to quantitate FOXP3+ cells under 40x magnification. How to define “hot-spot area”, only FOXP3+ cells in hot-spot area are counted? Or more than 5 or 10 fields are counted and then average the numbers? How to score IHC intensity (Figure to demonstrate weak, moderate, and strong, only >moderate expression should be counted).
Many studies use % x intensity = H score
2. Statistical analysis: can be more concise.
3. It is assumed FOXP3+ cells at diagnosis and post therapy are different. It has better separate patients into two groups to have comparison with FOXP3+ cells in control group, respectively.
Given different mechanisms between nodal FTH T-cell lymphoma and PTCL, NOS, it is assumed the FOXP3+ cells are different in the two subtypes of T-cell lymphoma. Did author have attempted to compare them at first?
4. It is also recommended to have an increase in sample size in control group, ideally to match with age and gender ratio.
Author Response
"Please see the attachment."

Reviewer 4 Report
Comments and Suggestions for Authors
Statistical Methods: High versus low FoxP3+ groups were defined by different cut-off values (125, 200 and 255, or using the cohort distribution for a value of above top tertile as shown previously) [6]. Nevertheless, the Cox proportional hazards model demonstrates that these cut-offs are not good for survival prediction. The key might be to explain why these points were chosen and what diverged in the results. Moreover, a thorough treatment of the implications for multivariate analysis results due to sample size constraints.
This study suggests there might be a non-linear effect (p=0.059) but concludes this is not statistically significant. Whether the non-linearity could arise with a larger cohort is an avenue I explore.
Sample Representation: Since the majority of our sample consists of data from Institute of Oncology Ljubljana this might restrict generalization. Highlighting the potential geographic or institutional biases that could exist due to differences in sample handling and immunohistochemistry, will elucidate the possible limitations of this study.
However, statistical power may be compromised after the subgroup analysis. The authors also note that the study was retrospective, but there is room for an explanation on how small subgroups can influence specific findings and circumscribe general conclusions.
While the paper does provide important information on comparing specific biomarker values across PTCL subtypes, there are limited datasets to compare with historical studies examining such markers (particularly extranodal and non-GATA positive cases), including a heterogeneity of both FoxP3+ counts and survival implications. There would be a better perspective on these results if those differences in question were acknowledged by also mentioning the opposite conclusions and whether they say more about the issue here since you selectively quote.
Clinical Utility: Although this study indicates the lack of FoxP3+ cells as an independent prognostic marker, its conclusion suggests that Treg function enhancing therapy trials should be recommended in PTCL. A bit of a stretch, really; Hard to see that in this study when the main outcomes show little or no real effect… If the analysis recommends combined therapeutic strategies in ongoing research, it must either be substantiated elsewhere by more substantial findings or labelled exploratory.
The conclusion about the need for larger, standardized studies is essential but could be expanded by discussing specific methodologies (e.g., prospective studies with standardized cut-off points and additional T-cell markers) that could strengthen future research. The recommendation for uniform FoxP3+ cell quantity cut-off values and units across studies is excellent, and the authors might consider including a proposed standard methodology to assist in future research validation.
Author Response
"Please see the attachment."

Round 2
Reviewer 1 Report
Comments and Suggestions for Authors
The authors put a lot of effort in answering my questions. This improved the manuscript.
Reviewer 3 Report
Comments and Suggestions for Authors
NA